# Investigation of the Protective Effect for GcMAF by a Glycosidase Inhibitor and the Glycan Structure of Gc Protein

**DOI:** 10.3390/molecules28041570

**Published:** 2023-02-06

**Authors:** Yoshimi Kanie, Yuya Maegawa, Yi Wei, Osamu Kanie

**Affiliations:** 1Research Promotion Division, Tokai University, 4-1-1 Kitakaname, Hiratsuka, Kanagawa 259-1292, Japan; 2Department of Applied Biochemistry, Tokai University, 4-1-1 Kitakaname, Hiratsuka, Kanagawa 259-1292, Japan; 3Graduate School of Science and Technology, Tokai University, 4-1-1 Kitakaname, Hiratsuka, Kanagawa 259-1292, Japan; 4Micro/Nano Technology Center, Tokai University, 4-1-1 Kitakaname, Hiratsuka, Kanagawa 259-1292, Japan

**Keywords:** iminocyclitol, α-GalNAc-ase, inhibitor, GcMAF, glycan structure

## Abstract

*O*-linked α-*N*-acetylgalactosamine (α-GalNAc) in the Gc protein is essential for macrophage activation; thus, the GalNAc-attached form of Gc protein is called Gc macrophage activating factor (GcMAF). *O*-linked glycans in Gc proteins from human plasma mainly consist of trisaccharides. GcMAF is produced when glycans on the Gc protein are hydrolyzed by α-Sia-ase and β-Gal-ase, leaving an α-GalNAc. Upon hydrolysis of α-GalNAc present on GcMAF, the protein loses the macrophage-activating effect. In contrast, our synthesized pyrrolidine-type iminocyclitol possessed strong in vitro α-GalNAc-ase inhibitory activity. In this study, we examined the protective effects of iminocyclitol against GcMAF via inhibition of α-GalNAc-ase activity. Detailed mass spectrometric analyses revealed the protective effect of the inhibitor on GcMAF. Furthermore, structural information regarding the glycosylation site and glycan structure was obtained using tandem mass spectrometric (MS/MS) analysis of the glycosylated peptides after tryptic digestion.

## 1. Introduction

The Gc (group-specific component) protein is a single-chain polypeptide with 458 amino acid residues and a molecular mass of approximately 51.2 kDa (UniProtKB database entry for VTDB_HUMAN: P02774) that belongs to the albumin family. There are 120 isoforms of the Gc protein in humans; however, most people carry one of the three main polymorphic forms. The difference between these polymorphic forms of Gc protein consists of point mutations at amino acid (a.a.) 416 (aspartic acid (D) or glutamic acid (E)) and a.a. 420 (lysine (K) or threonine (T)) in the peptide sequence [1,2,3,4,5,6] (Figure 1a). The plasma and serum of healthy individuals contain 75–450 mg/L of Gc protein. Gc protein is multifunctional and is also called Gc globulin and vitamin D-binding protein. The protein plays a role as a carrier for vitamin D and its plasma metabolites, an actin scavenger in plasma, and a factor in anti-angiogenesis and related antitumor activity [3,6,7,8].

This protein is post-translationally glycosylated and carries *O*-linked glycans consisting of α-*N*-acetylneuraminic acid (α-Sia), β-galactose (β-Gal), and α-*N*-acetylgalactosamine (α-GalNAc) attached to either ^418^Thr or ^420^Thr [1,3,4,5,6,9] (Figure 1b). Among these glycosylated variants, the Gc protein-carrying *O*-linked monosaccharide α-GalNAc activates macrophages; thus, the mono-glycosylated form of the Gc protein is called Gc macrophage activating factor (GcMAF) [3,6,10]. Trisaccharide-carrying proteins have been observed in approximately 1% of total Gc proteins. [1,4,6]. Sequential deglycosylation by α-Sia-ase, β-Gal-ase, and α-GalNAc-ase from the Gc protein leads to a loss of the activating effect on macrophages [11,12,13] (Figure 1c). Though the effect of GcMAF has been proven in the treatment of prognosis for patients with cancer and the concentration of nagalase (α-GalNAc-ase) in the patient’s blood has been used as a marker for cancer healing [14], no significant difference in the amounts of Gc protein-carrying trisaccharide in serum between healthy controls and patients with cancer has been reported [15]. The treatment of Gc protein with serum collected from patients with cancer or cell extracts containing α-GalNAc-ase derived from cancer cells reduces the activity of macrophages in vitro; however, the enzymological details at the molecular level remain to be revealed [14].

In contrast, iminosugars found in plants and micro-organisms have potential as glycosidase inhibitors [16,17,18,19,20]. Iminosugar with a pyrrolidine framework mimics the transition state of a substrate in the catalytic site of glycosidases, which often shows competitive inhibitory activity against glycosidases [21,22,23,24,25,26,27]. We discovered and reported a pyrrolidine-type specific and potent inhibitor, 2-hydroxymethyl-5-(phenethylaminomethyl)-pyrrolidine-3,4-diol, against α-GalNAc-ase from chicken liver with *K*i = 29 nM, with *p*-nitrophenyl (PNP)-α-GalNAc as a substrate at pH 4.0 [26]. We hypothesized that such an α-GalNAc-ase inhibitor could prevent the loss of GcMAF-activity. In this study, we investigated whether iminocyclitol prevents GcMAF hydrolysis by α-GalNAc-ase extracted from HepG2 cells. A difference in macrophage activity has been observed when adding GcMAF or Gc protein with or without treatment with HepG2 cellular extract to macrophages [12]. Considering the amount of glycosylated Gc protein with only a few percent of the total molecules, we decided to utilize a protocol that is often used in proteomics research to analyze picomole-scale proteins. This methodology allowed us to investigate the individual glycan structures attached to peptides, as well as the point-mutated glycosylated peptides. After tryptic digestion of the protein, glycopeptides were enriched using a carbon cartridge column and analyzed using liquid chromatography–mass spectrometry (LC–MS) [28,29,30]. Protective effects were evaluated by comparing the ratio of glycopeptides in target peptides that contain the same amino acid sequences.

## 2. Results and Discussion

### 2.1. Glycosidase Activities of a Cellular Extract from HepG2 Cells and Their Inhibition by an Iminocyclitol

The treatment of GcMAF with HepG2 cellular extract reduces the potency of macrophage activation [12]. HepG2 cells are adhesive cells derived from liver cancer cells in Caucasian males. HepG2-derived α-GalNAc-ase has a relatively higher activity than that derived from normal liver cell lines. The activities of α-GalNAc-ase, β-Gal-ase, and α-Sia-ase, which are involved in the hydrolysis of *O*-linked glycans in Gc protein, were investigated.

First, we examined the activity of glycosidases in the culture medium of HepG2 cells, which were found to be below the detection limits (unpublished data). The cells were homogenized and the obtained cellular extract was precipitated with 70% ammonium sulfate, following a method described by Mohamad, S. B. et al. [12]. They examined α-GalNAc-ase activities and determined exo-type α-GalNAc-ase activity from cultured HepG2 cells. The residue obtained was dissolved in a 50-mM citrate buffer at pH 5.0, 6.0, and 6.9. PNP- and 4-methylumberiferyl (4MU)-sugar derivatives were used as substrates for individual glycosidase reactions. α-GalNAc-ase and β-Gal-ase activities were observed in the extracts; however, α-Sia-ase activity was under the detection limit in our current experiments (Table 1). Higher activities of α-GalNAc-ase and β-Gal-ase were found at pH 5.0, consistent with their localization inside the cell [31].

One millimolar of PNP-α-GalNAc, PNP-β-Gal, and 0.2 mM of 4MU-α-Sia were used as substrates for α-GalNAc-ase, β-Gal-ase, and α-Sia-ase, respectively. The obtained protein concentration, determined by the bicinchoninic acid (BCA) method, was 89 mg/mL, 380 mg/mL, and 360 mg/mL at pH 5.0, 6.0, and 6.9, respectively.

Next, a time-course study of α-GalNAc-ase activity was performed at pH 5.0 (Figure 2a). The α-GalNAc-ase activity against PNP-α-GalNAc diminished to about half within 4 h (Appendix A). This rapid loss of enzyme activity was probably due to the stability of the enzyme and existence of contaminated proteases. Therefore, we used freshly prepared cell extracts by setting the reaction time to less than 3 h in the following experiments.

The iminocyclitol under investigation has been reported to be a potent inhibitor of α-GalNAc-ase derived from chicken livers. We examined its inhibitory activity against α-GalNAc-ase derived from HepG2 cellular extract and found it to be a potent inhibitor with an IC_50_ of 82 nM when PNP-α-GalNAc was used as a substrate (Figure 2b).

### 2.2. Analysis of O-Glycans in Gc Protein

Human plasma-derived Gc protein is commercially available, and it has been reported that a small proportion of the protein is *O*-glycosylated [4]. Thus, we decided to employ a proteomics methodology utilizing mass spectrometry (MS) in this study, which enabled us to analyze several hundred picomoles of glycans present in one mg protein. The cysteine residues in the Gc protein were reduced and alkylated before tryptic digestion. Glycosylated peptides obtained by trypsin treatment were enriched using a carbon cartridge column prior to LC–MS. The glycopeptide sequence was observed in the eluted fraction of 20% acetonitrile containing 0.1% formic acid. Because of the point mutations at positions 416 (D or E) and 420 (K or T) in the amino acid sequence (Figure 1a), the expected target amino acid sequences would be LPDATPTELAK and LPEATPTELAK for positions 414 to 424 and LPDATPK and LPEATPK for positions 414–420. In the current study, glycopeptides carrying the sequences of LPDATPTELAK and LPEATPTELAK were detected; however, glycopeptides carrying the sequences of LPDATPK and LPEATPK corresponding to a point mutation in the amino acid sequence at 420 could not be found. The expected mass numbers of each glycopeptide and peptide are shown in Figure 3a. The retention times of the target components were obtained from single-ion chromatograms (SIC) (Appendix A) by extracting the desired mass number from the total ion chromatogram (TIC) (Figure 3b). The target glycopeptides were observed at retention time from 17.0 min to 17.9 min (area 1) and from 24.2 min to 26.1 min (area 2). The mass numbers of HexNAc (GalNAc)-carrying peptides with a sequence of LPD/EATPTELAK were observed at *m/z* 679.5 and 686.5 in area 1 (Figure 3c), and HexNAc (GalNAc)-, Hex (Gal)-HexNAc (GalNAc)-, and Sia-Hex (Gal)-HexNAc (GalNAc)-carrying peptides with the same sequences were observed at *m/z* 679.5 and 686.5, *m/z* 760.5 and 767.5, and *m/z* 906.3 and 913.3, respectively, in area 2 (Figure 3c,d). Notably, monosaccharide-carrying peptides were observed in two separate areas, as explained in Section 2.4. These glycopeptides were confirmed to contain protonated divalent ions. The mass spectra of peptides without glycans were observed at approximately 29 min (Figure 3b). Peptide signals are often detected as multivalent ions in electrospray ionization mass spectrometry (ESI-MS) when the molecular weight is higher than 1000. All signals related to glycopeptides were observed in a pair of peptides with the amino acid sequences of LPDATPTELAK and LPEATPTELAK. Although detailed glycan structures, including linkage position, anomericity, and homogeneity, have not yet been determined, information was not necessary for tracing mixed enzymatic reactions.

### 2.3. Protective Effect of Iminocyclitol for GcMAF

Having confirmed the potent inhibitory effect of iminocyclitol on α-GalNAc-ase derived from HepG2 cell extract and the retention times of glycosylated peptide in high-performance liquid chromatography (HPLC) together with mass numbers of individual ions in focus, we next examined whether iminocyclitol inhibits α-GalNAc-ase, which protects the glycosylated Gc protein, GcMAF, from hydrolysis. Figure 4a shows an overview of our investigation of the protective effect of the Gc protein. We focused on the following questions: (1) can iminocyclitol inhibit α-GalNAc-ase derived from HepG2 cells using glycopeptides obtained by tryptic digestion of Gc protein? (2) can the Gc protein be a substrate of α-GalNAc-ase derived from HepG2 cells? (3) can we confirm the protective effect of iminocyclitol against α-GalNAc-ase derived from HepG2 cells using Gc proteins as a substrate? Since HepG2 did not have α-Sia-ase activity, as shown in Section 2.1, we decided to use α-Sia-ase derived from *Vibrio cholera*, which cleaves both αSia-(2→3)- and -(2→6)-βGal linkages [32], in addition to HepG2 cellular extract.

#### 2.3.1. Glycosidase Treatment of the Tryptic Digests of Gc Protein and the Effect of an Iminocyclitol

First, the glycopeptide fraction obtained by tryptic digestion of the Gc protein was treated sequentially with α-Sia-ase and HepG2 cellular extract (Figure 4a). The samples were then subjected to LC–MS. Note that the quantitative analysis was not possible in our current analysis setup based on the ESI–ion trap MS. Thus, areas for ions of individual compounds in a mass chromatogram were compared. Our major concern is also with regards to handling errors in the enrichment process of glycopeptides during experiments. Instead, we focused on the intensity differences of glycosylated peptides in an individual experiment to see the tendency of changes. (See Appendix A for similar experimental data) The quantities of each glycopeptide under a series of reaction conditions were compared for each peptide sequence separately (LPD/EATPTELAK) (Figure 4b). The effect of glycosidases was apparent in the comparison of a control experiment with an enzyme-treated one, wherein the intensities of all glycopeptides, mono-, di-, and tri-saccharide-carrying peptides were decreased. The inhibitory effect was confirmed by comparing the glycopeptide intensities in the experiments with and without the inhibitor. Therefore, we concluded that glycopeptides serve as substrates of glycosidases and that iminocyclitol inhibits enzyme activity. An averaged mass spectrum was obtained for the individual ions that are the focus of the SIC obtained from the TIC; furthermore, the sum of the averaged intensities over time of elution in chromatography was used for comparison. Although the ion intensities in mass spectrometry are not quantitative, because of the ionizability of individual molecules, the intensities can be used as a guide for the increase or decrease of each component.

#### 2.3.2. Inhibition of Glycosidases by an Iminocyclitol Using Gc Protein as a Substrate

Next, we focused on glycan hydrolysis and its inhibition using the Gc protein (Figure 4a). The Gc protein was first treated with the enzyme mixture described earlier, followed by digestion with trypsin. Since we found that the glycosidase activities of HepG2 cellular extract decreased rapidly, probably due to the contaminated proteases, we initially examined an enzyme mixture of α-Sia-ase from *Vibrio cholerae*, β-Gal-ase from *Aspergillus oryzae*, and α-GalNAc-ase from chicken liver as a positive control. The glycopeptides obtained were analyzed based on the above method. The effect of glycosidase on Gc protein was confirmed and the inhibitory activity of iminocyclitol against α-GalNAc-ase was also observed (Figure 4c). Having confirmed the changes in the areas reflecting the amount of glycopeptides as a result of glycosidase activities using Gc protein as a substrate, the effect of iminocyclitol was then evaluated in experiments using HepG2 glycosidases (Figure 4d). As the Gc protein is known to carry not only vitamin D but also actin fragments and fatty acids, we added oleic acid in these experiments [33]. As a result, the areas of all glycopeptides carrying mono-, di-, and trisaccharide in the chromatogram increased, though the changes were not significant. This result is puzzling because the trisaccharide portion was increased, as was the case for the earlier experiments using glycopeptides as substrates. We suspect the following two reasons for the observed low protective effect when the HepG2-derived α-GalNAc-ase was used. (1) IC_50_ value of the iminocyclitol against HepG2-α-GalNAc-ase was relatively high compared with that against α-GalNAc-ase from chicken liver, which might have affected the protective effect. (2) The relative velocity of α-GalNAc-ase might have decreased when Gc protein was used as the substrate. The fact that the intensity of trisaccharide became higher when the 14 μM of iminocyclitol were used suggests that the iminocyclitol might inhibit the α-Sia-ase from *Vibrio cholerae*. We thus examined the inhibitory effect using 0.2 mM MU-α-Sia as a substrate and found that the iminocyclitol inhibited the α-Sia-ase with IC_50_ = 18 μM. The inhibitory effect was 3% and 40% in the case of the 1 μM and 14 μM inhibitors, respectively. Regarding the effect against β-Gal-ase, 23% inhibition was observed at 10 μM for the enzyme from *Aspergillus oryzae* [25]. Therefore, it is conceivable that the used iminocyclitol inhibited β-Gal-ase and α-Sia-ase at higher concentrations. Regardless of the inhibitory specificity, a small increase of GalNAc-peptide suggests that the iminocyclitol did inhibit its hydrolysis and protected GcMAF from α-GalNAc-ace. The enzymes on B and T cells are involved in producing GcMAF from the Gc protein [34]. At the site of inflammation, B cells and T cells deglycosylate *O*-glycan on the Gc protein, leading to GcMAF. If the inhibitor could be introduced to the site of inflammation in advance using a liposome or other drug, it would result in the protection of GcMAF.

### 2.4. Analyses of Structures of O-Glycans and Their Site in Gc-Protein

Analyses of the extracted chromatograms corresponding to the individual ions of interest from the TIC provided retention times for individual entities. Figure 5a shows the extracted SIC of each *m/z* value of monosaccharide (blue)-, disaccharide (green)-, and trisaccharide (red)-linked peptides. Notably, the peak of trisaccharide-carrying glycopeptide eluted at 13 to 16 min was originally observed as a single peak on the chromatogram before enzymatic treatment; however, the peaks of trisaccharide-linked peptides were separated into two peaks after treatment with α-Sia-ase and HepG2 cellular extract. The mass numbers corresponding to the glycopeptide with trisaccharide at *m/z* 906.3 (peak 1 and peak 2) for LPDATPTELAK and *m/z* 913.3 (peak 3 and peak 4) for LPEATPTELAK were further analyzed using tandem mass spectrometry (MS/MS). The obtained fragments were mainly produced by the cleavage of the glycosyl linkages (Figure 5b). A fragment ion that lost sialic acid was detected predominantly, and a sequence corresponding to Sia-HexNAc was not observed. Furthermore, energy-resolved mass spectrometry (ERMS) of these ions revealed different energy dependencies for decay (Appendix A) [35]. These results suggest that there are two glycosylation sites in the peptide, where the same glycan is attached to one of these sites only.

To our surprise, the *N*-acetylated hexosamine (HexAc)-attached peptides with *m/z* 679.5 (LPDATPTELAK) and *m/z* 686.5 (LPEATPTELAK), and disaccharide-attached peptides with *m/z* 760.5 and *m/z* 767.5 were observed in four separate peaks, respectively. At this point, we consider possible reasons for this phenomenon. Focusing on the peptide sequence, a proline next to threonine that carries *O*-glycan was found. A partial structure of nephritogenoside, namely tri-glucoside attached to threonine next to proline, showed two isolated peaks in HPLC, which were caused by the *cis*- and *trans*-isomerism of proline amide [36]. Simple cases have also been reported for peptides [37,38]. In any case, the isomeric structures were in pH-dependent equilibrium. Based on these considerations, we suggest that glycopeptides carrying GalNAc at ^418^Thr or ^420^Thr may also exist in the *cis*- and *trans*-isomers associated with proline. In the case of trisaccharides, the presence of a negative charge in sialic acid diminished the separation of isomers in the chromatogram. Furthermore, the large difference in the elution time for the disaccharide can be explained by the added steric bulk of the glycan structure.

The glycosylation site could be determined by alkaline hydrolysis of *O*-glycan to give a peptide with a mass of a.a.–18, equivalent to a dehydrated amino acid by β-elimination of glycan [39]. From the MS analysis, the observed *m/z* values were assigned to the dehydrated peptide sequences of LPEATPTELAK(–H_2_O) and LPDATPTELAK(–H_2_O) at *m/z* 1137.4 and *m/z* 1151.4, respectively (Figure 6a) in the positive ion mode. These signals were further examined using MS/MS analysis. Both the precursor ions with *m/z* 1137.4 and *m/z* 1151.4 produced ion with *m/z* 640.2 corresponding to the y-ion of dehydrated PTELAK [1,3,4] (Figure 6b,c). The ion with *m/z* 512.1 was only observed in the MS/MS analysis of the precursor ion with *m/z* 1151.4, which was associated with the b-ion for dehydrated LPEAT (Figure 6c).

## 3. Materials and Methods

### 3.1. Chemicals

Gc protein (Human plasma) was purchased from Athens Research & Technology Inc. (GA, USA). RapiGest SF Surfactant was obtained from Waters Co., Ltd. (Milford, MA, USA). Dithiothreitol (DTT) and 2-iodoacetamide were obtained from Nacalai Tesque (Kyoto, Japan). *p*-Nitrophenyl 2-acetamido-2-deoxy-α-D-galactopyranoside (PNP-α-GalNAc), *p*-nitrophenyl β-D-galactopyranoside (PNP-β-Gal), 2′-(4-methylumbelliferyl)-α-D-*N*-acetyl neuraminic acid sodium salt hydrate (4MU-α-NeuAc), *p*-nitrophenol (PNP-ol), α-neuraminidase (Sia-ase) from *Vibrio cholerae* (EC 3.2.1.18), β-galactosidase (β-Gal-ase) from *Aspergillus oryzae* (EC 3.2.1.23), α-*N*-acetylgalactosaminidase from the chicken liver (EC 3.2.1.49), and Penicillin-streptomycin, cOmplete^TM^ protease inhibitor cocktail tablets, ammonium hydrogen carbonate, and trifluoroacetic acid were obtained from Sigma-Aldrich Co. (St. Louis, MO, USA). 4-Methylumbelliferone, mass spectrometry grade trypsin (EC 3.4.21.4) from porcine pancreas, D-MEM (containing low glucose and L-glutamate), PBS (–), ethylenediaminetetraacetic acid (EDTA), tris (hydroxymethyl) aminomethane, ammonium sulfate, and HCl were obtained from Wako Pure Chem. (Osaka, Japan). Fetal bovine serum was purchased from Gibco, Thermo Fisher Scientific (Waltham, MA, USA). Spectra/Por7 dialysis membrane, 12 mm (flat widths), MWCO15000 were purchased from Spectrum Laboratories, Inc. (Rancho Dominguez, CA, USA). BCA Protein Assay Kit was obtained from TaKaRa Bio Co., Ltd. (Shiga, Japan). Carbograph column was obtained from Alltech Associates Inc. (Chicago, IL, USA). HepG2 cells (Hepatocellular carcinoma cells, human) were purchased from Cosmo Bio Co., Ltd. (Tokyo, Japan). LC grade solvents and MilliQ water were used throughout the experiments.

### 3.2. LC-MS Analysis of Glycopeptides

Glycopeptide mixtures were separated by the Prominence HPLC system (Shimazu, Kyoto, Japan) using a PEGASIL ODS 3 μm 1 mm i.d. × 100 mm column, and by the mobile phase of 0.1% formic acid (FA) and acetonitrile containing 0.1% FA with linear gradient mode at room temperature. The flow rate was 50 μL/min, and the detection was performed on an Esquire 3000 plus mass spectrometer equipped with an electrospray ionization interface (Bruker Daltonics GmbH, Bremen, Germany). The mass spectrometric parameters were as follows: polarity = positive ion mode, ion source gas (nitrogen) pressure = 10 psi, dry gas = 4.0 L/min, dry temperature = 250 °C and scan range = from 400 to 2000 *m/z*. In MS/MS experiments, the precursor isolation window was set to ±2 Da, and He was used as the collision gas. The operation was carried out on trapControl software and data were acquired and processed with Compass DataAnalysis (Bruker Daltonics GmbH, Bremen, Germany).

### 3.3. Cell Culture and Preparation of Cellular Extract

HepG2 cells were grown to 90% confluence (1.9 × 10^6^ cells) in D-MEM culture medium, containing 10% (*v*/*v*) heat-inactivated fetal calf serum, in the presence of 1% (*v*/*v*) penicillin-streptomycin on a dish coated with collagen (Cellmatrix I) at 37 °C with 5.0% CO_2_. The medium was replaced every day. Cells were collected by scraping with 0.1% EDTA/PBS then 15 mM Tris/HCl buffer (pH 7.0, 3 mL) and glass beads (diameter <106 μm, 50 mg) were added. The cell suspension was sonicated for 3 min at 3 intervals on ice using a Sonicator Ultrasonic Processor XL (Misonix, New York, NY, USA) followed by ultracentrifugation (11,000*g*) for 15 min at 4 °C. The supernatant was collected and dialyzed in 70% ammonium sulfate overnight at 4 °C. Then, the precipitate was collected by removing the supernatant after centrifugation (11,000*g*) for 20 min at 4 °C. The residue was dissolved in 2 mL of 50 mM citrate buffer (pH 5.0, 6.0, 6.9) and dialyzed in 50 mM citrate buffer (pH 5.0, 6.0, 6.9) for 6 h at 4 °C. Finally, the obtained solution was brought to an amount of 2 mL by adding 50 mM citrate buffer (pH 5.0, 6.0, 6.9).

### 3.4. Glycosidase Activity in the Cellular Extract from HepG2

The amount of protein in the cellular extract was determined using TaKaRa BCA Protein Assay Kit. Bovine serum albumin (BSA) solution was used as a standard solution (5–200 μg/mL). Following the instructions, the assay reagent was prepared and the assay was carried out in a total volume of 200 μL for 1 h at 60 °C. The UV/vis absorbances of the reaction solutions were measured at 560 nm for the reaction products of the chelation of two molecules of bicinchoninic acid (BCA) with one cuprous cation (Cu^+^), using a microplate reader (SH-9500 Lab, Corona electric, Ibaraki, Japan).

To determine glycosidase activity, all reactions were performed in a total volume of 100 μL. The substrates used were PNP-α-GalNAc, PNP-β-Gal, and 4MU-NeuAc for the evaluation of each glycosidase activity. The reaction mixtures contained each substrate (1 mM PNP-α-GalNAc, 1 mM PNP-β-Gal, 0.2 mM 4MU-α-NeuAc) and the cellular extracts from HepG2 in 50 mM citrate buffer (pH 5.0, 6.0, 6.9) were incubated for 3 h at 37 °C. The reaction was terminated by the addition of 500 mM Na_2_CO_3_ (pH 12.5) of 100 μL. The reaction solutions were measured, under basic conditions over pH 11.6 by UV/vis absorbance at 415 nm for PNP-ol and fluorescence at 447 nm (excitation wavelength at 325 nm) for 4-methylumbelliferone, using a microplate reader (SH-9500Lab). Further examinations of α-GalNA-ase were performed in various incubation times at pH 5.0, and inhibitory activity was evaluated for 3 h incubation at 37 °C.

### 3.5. Preparation of Glycopeptides from Gc Protein

The amounts of 160 μL of 50 mM NH_4_HCO_3_ (pH 8.0) and of 6 μL of 4% (*w*/*v*) RapiGest in water (final concentration of 0.1%) were added to 0.2 mg Gc protein in 40 μL water. This mixture was incubated for 10 min at 40 °C, dithiothreitol was added at a final concentration of 5 mM, and incubated for 15 min at 80 °C. Iodoacetamide was added at a final concentration of 15 mM and reacted in the dark at room temperature for 30 min. Tryptic digestion was preceded by the addition of trypsin (final trypsin/ protein ratio of 1/50 (*w*/*w*)) and incubating overnight at 37 °C. The digestion was terminated by the addition of trifluoroacetic acid (TFA) and acetonitrile at the final concentrations of 1% each. To concentrate glycopeptide, the tryptic digest was applied to a Carbograph cartridge column. The glycopeptide mixture was eluted using 4 mL of 20% acetonitrile containing 0.1% TFA and concentrated.

### 3.6. The Inhibition Study of an Iminocyclitol against Digestion of Glycans by α-Sia-ase and HepG2 Cellular Extract Using Glycopeptides

The glycopeptides (20% eluent fraction from Carbograph cartridge) were concentrated and α-Sia-ase (*Vibrio cholerae*), HepG2 cellular extract with/without 1.0 μM iminocyclitol were added in 50 mM citrate buffer (pH 5.0) containing 5 mM CaCl_2_. Then the reaction mixture was incubated at 37 °C overnight. Finally, the reaction mixtures were analyzed by LC–MS.

### 3.7. Treatment of Gc Protein with Glycosidases and Evaluation of the Inhibitory Activity of Iminocyclitol

Incubations were performed in a total volume of 200 μL. The mixtures of 0.1 mg of Gc protein and the mixture of α-Sia-ase (*Vibrio cholerae*), β-Gal-ase (*Aspergillus oryzae*) and α-GalNAc-ase (chicken liver) were incubated with and without 1.0 μM iminocyclitol in 50 mM acetate buffer (pH 6.0) containing 5 mM CaCl_2_ at 37 °C overnight. Then, the reaction mixture was treated following the above protocol (see 3.5.). In the experiment with the HepG2 cellular extract, following the treatment of 0.1 mg Gc protein with 0.5 mU α-Sia-ase (*V. cholerae*) in a 50 mM citrate buffer (pH 5.0) containing 5 mM CaCl_2_ for 1.0 h at 37 °C—which itself followed incubation with three equivalent moles of oleic acid in 50 mM citrate buffer (pH 5.0) at 4 °C overnight—the reaction mixture was incubated with HepG2 cellular extract (7 μg protein) with or without inhibitor (4 μM) for 2.0 h at 37 °C.

### 3.8. β- Elimination of O-Glycan from Glycopeptides

The concentrated glycopeptides were dissolved in 50 μL of 0.1 M NaOH. After the reaction mixture was incubated overnight at 57 °C, Dowex 50WX8 was added and stirred for 1.0 h. The supernatant was applied on ZipTipC18 and eluted with 80% acetonitrile containing 0.1% TFA. The obtained solution was directly analyzed by mass spectrometry.

## 4. Conclusions

The effect of synthetic iminosugar with a pyrrolidine scaffold—in which the C-2(*R*), C-3(*R*), C-4(*S*), and C-5(*R*)-configurations are equipped with a phenethylamino group at the C-1 position—on Gc protein treated with a HepG2 cellular extract was investigated using proteomics method. HepG2 cells derived from human liver cancer cells were chosen as the source of glycosidases in this study. The cell lysate contained critical activities of α-GalNAc-ase and β-Gal-ase. However, α-sialidase activity could not be observed, probably due to the low concentration of the enzyme. In the current investigation, it was shown that: (1) an inhibitory activity of iminocyclitol in terms of several tens of nmol/mg/min was confirmed against α-GalNAc-ase using PNP-α-GalNAc as a substrate; (2) α-GalNAc group of GcMAF, which was obtained by the treatment of Gc protein with an extract of HepG2 containing β-Gal-ase and α-GalNAc-ase and an externally added α-Sia-ase, was hydrolyzed; and (3) the protective effect of the iminocyclitol for GcMAF was observed to some extent, though we could not observe a clear protective effect due to the unexpectedly discovered weak inhibitory effects against α-sialidase. Furthermore, we confirmed that the main component of the glycan structure in the Gc protein was a linear trisaccharide consisting of α-Sia, β-Gal, and α-GalNAc. Regarding the glycosylation site, only one of the two Thr residues was glycosylated.

## Figures and Tables

**Figure 1 molecules-28-01570-f001:**
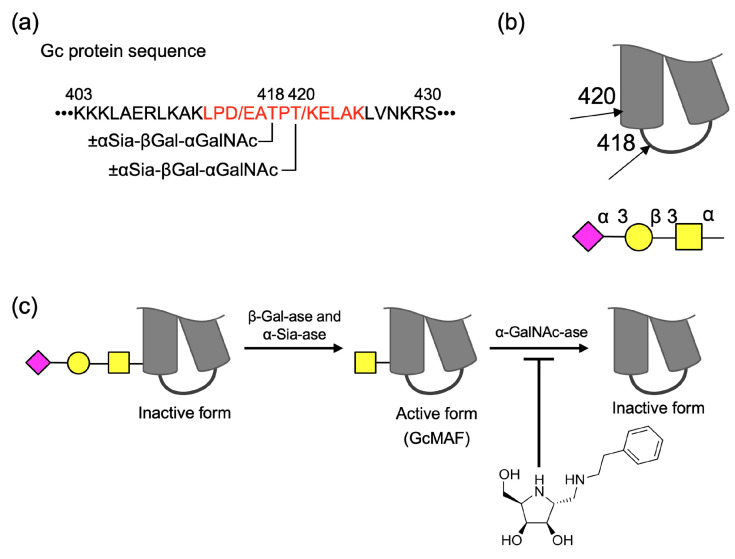
Partial structure of the Gc protein related to the investigation and research concept. (**a**) A sequence from a.a. 403 to 430 that forms the helix-turn-helix in the Gc protein is shown and an expected sequence after tryptic digestion is shown in red. Site of glycosylation is provided with glycan structure. (**b**) Glycosylation sites at ^418^Thr and ^420^Thr located near the turn structure are indicated by arrows. (**c**) After hydrolyses of sialic acid and galactose, *N*-acetylgalactosamine-attached Gc protein can activate macrophage; however, hydrolysis of the monosaccharide leads to the loss of its activity. Synthetic iminocyclitol might protect the GcMAF. Symbol nomenclature was used according to the guideline of Society for Glycobiology (SFG).

**Figure 2 molecules-28-01570-f002:**
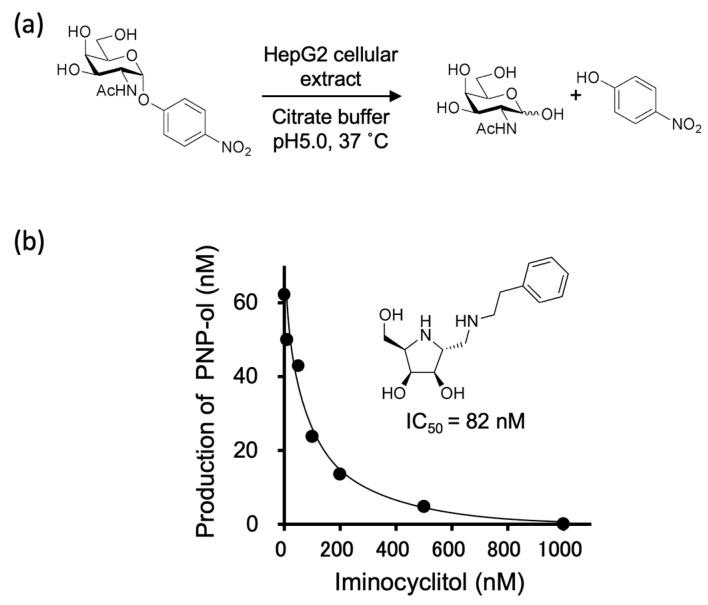
(**a**) Cell extract from HepG2 containing α-GalNAc-ase was evaluated using PNP-α-GalNAc as a substrate. (**b**) Inhibitory activity of the iminocyclitol against α-GalNAc-ase from HepG2 cellular extract. PNP-α-GalNAc(1 mM) and the cellular extract were incubated for 3 h at 37 °C in a 50-mM citrate buffer (pH 5.0). 2-Hydroxymethyl-5-(phenethylaminomethyl)-pyrrolidine-3,4-diol is a synthetic pyrrolidine type iminocyclitol with C-2(*R*), 3C(*R*), C-4(*S*), C-5(*R*)-configurations.

**Figure 3 molecules-28-01570-f003:**
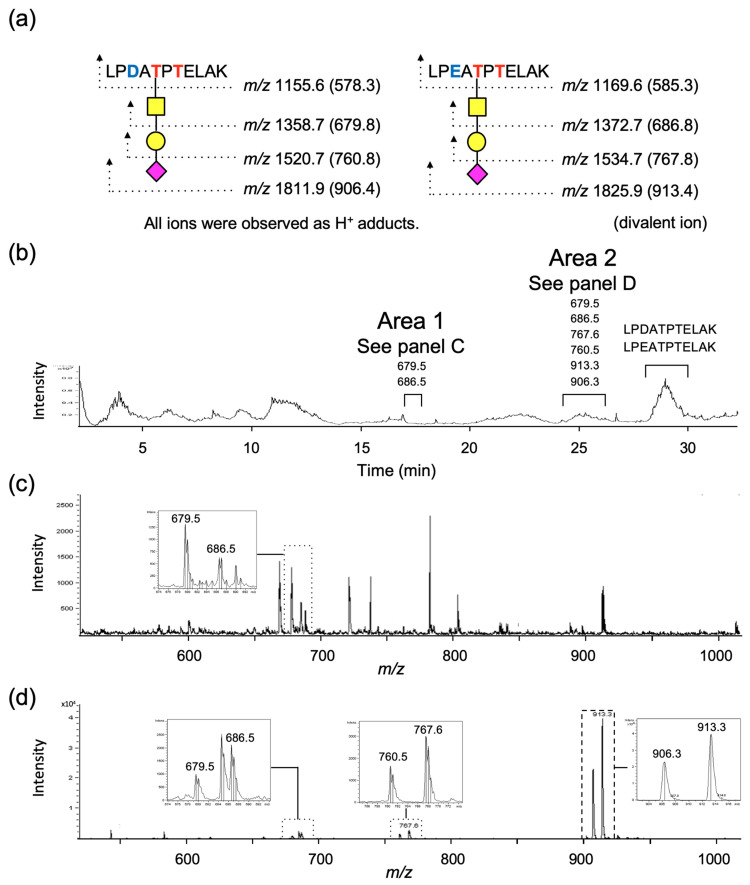
(**a**) Expected mass numbers of protonated glycopeptides and peptides for tryptic digests. (**b**) LC–MS analysis of Gc protein after tryptic digestion; total ion chromatogram in positive ion mode where glycopeptides and peptides of interest are shown. (**c**) Mass spectrum of area 1 (17 min to 17.9 min) showing presence of GalNAc-carrying peptides. (**d**) Mass spectrum of area 2 (24.2 min to 26.1 min) consisting of mono-, di-, and trisaccharide-carrying peptides.

**Figure 4 molecules-28-01570-f004:**
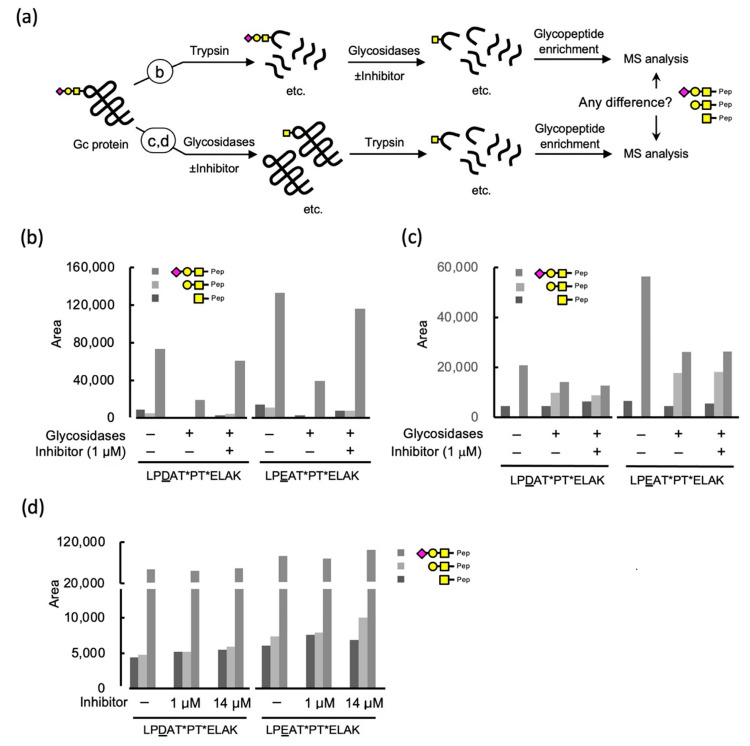
(**a**) Experimental setups for the evaluation of glycan hydrolysis and its inhibition before and after Trypsin treatment, relying on MS-based proteomics methodology. Areas for individual glycopeptides associated with the major isoforms LPDATPTELAK and LPEATPTELAK are shown. (**b**) Effects of glycosidases and GalNAc-inhibitor on the composition and intensities of peptides and glycopeptides were evaluated after tryptic digestion followed by glycosidase treatment (±). Glycosidase treatment decreased the glycans, which were inhibited by the used inhibitor. (**c**) Effects of glycosidases and GalNAc-inhibitor on glycopeptides intensities after ±glycosidase treatment, followed by tryptic digestion. (**d**) Effects of different concentrations of inhibitor on composition and areas of glycopeptides after ±glycosidases (α-Sia-ase and HepG2 cellular extract) followed by tryptic digestion.

**Figure 5 molecules-28-01570-f005:**
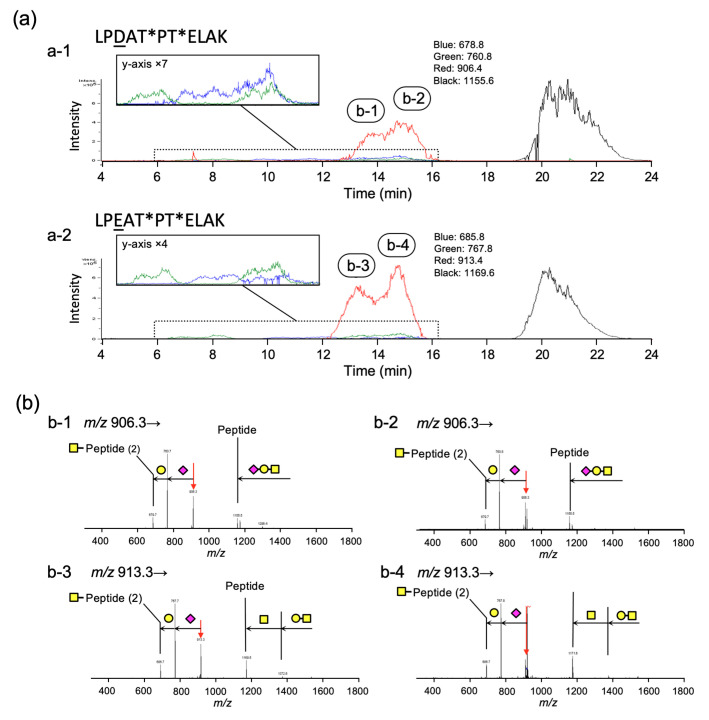
(**a**) Extracted chromatogram of glycopeptides after α-Sia-ase and HepG2 cellular extract treatment on Gc-protein. (a-1) *m/z* 678.8 (monosaccharide-attached peptide): blue; *m/z* 760.8 (disaccharide-attached peptide): green; and *m/z* 906.4 (trisaccharide-attached peptide): red. (a-2) *m/z* 685.8 (monosaccharide-attached peptide): blue; *m/z* 767.8 (disaccharide-attached peptide): green; and *m/z* 913.4 (trisaccharide-attached peptide): red. (**b**) MS/MS spectra of glycopeptides carrying trisaccharide with its peptide sequence of LPDATPTELAK and LPEATPTELAK. (b-1) Precursor ion: *m/z* 906.3 in the peak around 13.9 min with its peptide sequence of LPDATPTELAK. (b-2) Precursor ion: *m/z* 906.3 in the peak around 14.8 min with its peptide sequence of LPDATPTELAK. (b-3) Precursor ion: *m/z* 913.3 in the peak around 13.5 min with its peptide sequence of LPEATPTELAK. (b-4) Precursor ion: *m/z* 913.3 in the peak around 14.6 min with its peptide sequence of LPEATPTELAK. *: possible site of glycan attachment.

**Figure 6 molecules-28-01570-f006:**
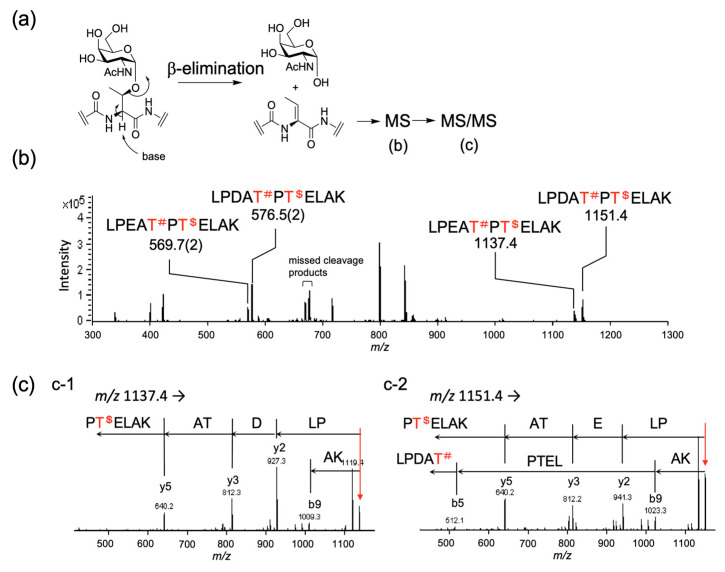
(**a**) Schematic presentation of the determining of the site of glycosylation by β-elimination followed by MS and MS/MS analysis. (**b**) MS spectrum of peptides after β-elimination of glycans. Possible glycosylation sites are indicated as T^#^ and T^$^. (**c**) MS/MS spectra for precursor ions *m/z* = 1137.4 and 1151.4 in panel B. (c-1) The MS/MS spectrum of ion with *m/z* 1137.4 reveals that ^420^Thr is the sole site of glycosylation. (c-2) The MS/MS spectrum of ion with *m/z* 1151.4 indicates that both ^418^Thr and ^420^Thr were glycosylated.

**Table 1 molecules-28-01570-t001:** Glycosidase activities of extracts from HepG2 cells.

	pH		
	5.0	6.0	6.9
α-GalNAc-ase	1.07	0.21	0.31
β-Gal-ase	2.00	0.25	0.36
α-Sia-ase	<0.01	<0.01	<0.01

Unit: nmol/mg (total protein)/min.

## Data Availability

Not applicable.

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
