# Peer review of "Investigation of the Protective Effect for GcMAF by a Glycosidase Inhibitor and the Glycan Structure of Gc Protein"

_molecules, 2023, doi:10.3390/molecules28041570_

Round 1
Reviewer 1 Report
In this manuscript, Kanie et al. examined the protective effects of synthetic iminocyclitol against Gc macrophage activating factor (GcMAF) via inhibition of α-GalNAc-ase activity. Detailed mass spectrometric analyses revealed the protective effect of the inhibitor on GcMAF. Furthermore, structural information regarding the glycosylation site and glycan structure was obtained using tandem mass spectrometric (MS/MS) analysis of the glycosylated peptides after tryptic digestion.
GcMAF is produced from Gc that has trisaccharides upon hydrolysis of sialic acid and galactose by α-Sia-ase and β-Gal-ase in human plasma. O-Linked α-GalNAc in the Gc protein is essential for macrophage activation. Upon hydrolysis of α-GalNAc, Gc protein loses the macrophage-activating effect. This study found that synthesized pyrrolidine-type iminocyclitol possessed strong in vitro α-GalNAc-ase inhibitory activity. Overall, the research are well designed with great rational and their data support the conclusion. The manuscript is well written.
Just one concern is the sialidase activity. They found that the extract from HepG2 cells did not have sialidase activity. It is hard to understand. There might be two reasons on why they did not observe sialidase activity in their research. One might be the enzyme activity assay condition that could not measure the enzyme activity. Another reason might be the cell extraction preparation condition that destroyed or denatured the sialidase enzyme. In this manuscript, they used 4MU-α-Sia as substrate to measure the sialidase activity, which measures the fluorescent intensity of the released 4MU. Although sialidase prefers acidic condition, it should be noticed that measuring fluorescent intensity of free 4-MU needs in a basic condition, like pH 10.4. They used 500 mM to terminate the enzyme reaction, but not sure the final what pH value of the sample solution is. In general, 0.25M glycine-NaOH (pH 10.4) (x mL) is used to stop the reaction (Feng C, Stamatos NM, Dragan AI, Medvedev A, Whitford M, Zhang L, Song C, Rallabhandi P, Cole L, Nhu QM, Vogel SN, Geddes CD, Cross AS. Sialyl residues modulate LPS-mediated signaling through the Toll-like receptor 4 complex. PLoS One. 2012;7(4):e32359. doi: 10.1371/journal.pone.0032359. Epub 2012 Apr 9. PMID: 22496731; PMCID: PMC3322133.). So, I suggest that they check on this issue. In addition, they can use other substrate such as GM3 to measure the sialidase activity as well. Second, I suggest that they use western blot to check the expression level of sialidase, which will show if the endogenous sialidase is involved in sialic acid removal from Gc protein.
A suggestion on literatures, please provide some literatures related to discussion “The enzymes on 222 B and T cells are involved in producing GcMAF from the Gc protein.” On page 7, first paragraph.
Author Response
The authors thank to this reviewer pointing out important issues. We addressed the issues and improved our manuscript.
Regarding the concerns about the sialidase activities, the authors would like the reviewer to understand first that the particular enzyme activity is not the main issue in the investigation. However, we realize the importance of the right conditions of the enzyme reactions, and included the detailed conditions. We updated the Materials and method section 3.4 (line 364–372) in order to make the above detail clear.
Secondly, about the extraction method of enzymes, we followed a method described by Mohamad et al. (newly added reference 12), which is added in the revised manuscript (line 96–104). The paper described the extraction of endogenous glycosidases from cultured HepG2 cells, however, GalNAc-ase among extracted enzymes only was used to cleave GalNAc-Gc protein (GcMAF) in the paper. Beforehand of this treatment, they treated the Gc protein with exogenously added Sia-ase and Gal-ase to convert Gc protein into MAF. Thus they focused on tumor-associated GalNAc-ase only. We suspected that the HepG2 cells produced Sia-ase and Gal-ase as well. Therefore, we attempted to extract enzymes supposedly containing three enzymes. However, the Sia-ase activity could not be observed in our hands, although we did find the Gal-ase activity. It might be that the activity of Sia-ase is below the detection limit in our current experiments. Regarding the suggestion of using GM3 as a substrate, we decided not to perform the extra experiment considering the fact that fluorogenic substrate could not be detected.
Osamu Kanie
Reviewer 2 Report
In this study, the authors attempted to examine the protective effects of iminocyclitol against GcMAF via inhibition of α-GalNAc-ase activity. The authors mentioned that they focused on three questions: (1) can iminocyclitol inhibit α-GalNAc-ase derived from HepG2 cells using glycopeptides obtained by tryptic digestion of Gc protein? (2) can the Gc protein be a substrate of α-GalNAc-ase derived from HepG2 cells? (3) can they confirm the protective effect of iminocyclitol against α-GalNAc-ase derived from HepG2 cells using Gc proteins as a substrate? However, evidence based answers to above questions were not clearly given.
1) Fig 3, selected ion chromatograms (SICs) for each glycopeptides should be given to show their elution peaks (at least given in the complementary material).
2) Figure 4. (b,c) the number of replicates should be given, statistic analysis should be performed. The bars corresponding to which glycopeptides should be indicated.
3) Line 220-222, “Although we have no explanation for this phenomenon, a small increase in the GalNAc-peptide indicates that iminocyclitol inhibits its hydrolysis and protects it from α-GalNAc-ace”. This is not a sound conclusion.
4) Key data and results should be clearly given to answer their questions and data based conclusion should be drawn in the end.
Author Response
Thanks to the reviewer. We revised our manuscript taking the valuable suggestions into account.
- We added SICs of individual glycopeptides and peptide (new Fig. 2S) as suggested.
- Quantitative analysis was not possible in our setup based on the ESI-iontrap MS. Furthermore, inaccurate mass intensities over different experiments and handling error associated with the enrichment process of glycopeptides were our major concerns. Instead, we focused on the intensity difference in an experiment to see the tendency of changes. The bars of graphs are now indicated. Also, responding to other reviewers’ comments, we updated Fig. 4.
- Additional experimental evidence for the inhibitory effect of the iminocyclitol against sialidase and discussion were included. (line 226–240)
- Regarding the questions we raised and the conclusion, we revised the conclusion section to make more clear (line 417–425).
Osamu Kanie
Reviewer 3 Report
The work by Kanie et al. describes the protective effect of the synthetic iminocyclitol inhibitor over the hydrolysis of the Thr-linked GalNAc monosccharide to the Gc protein.
The authors make use of tandem mass spectrometry (MS/MS) to profile the site specific and glycosylation content of the Gc protein previously treated with a cocktail of glycosidases from the cellular extract from HepG2 cells and compare the results in presence and absence of the synthetic inhibitor. The study is very relevant since the a-GalNAc on the Gc protein confers a macrophage-activating effect.
However, a key point of the data analysis requires discussion. The authors evaluate the inhibition of glycosidases by iminocyclitol using Gc protein as a substrate. Section 2.3.2. In figure 4b, the plot-bar shows the areas of the different glycoforms in presence and absence of glycosydases and inhibitor. In (b) it is questionable how in presence of glycosydases but not inhibitor the content of trisacchide-linked Gc protein is reduced while in presence of both, glycosidases and the inhibitor, this form, but also the disaccharide and monosaccharidic forms, are higher. It looks like the iminocyclitol has an inhibitory effect also on sialidase and b-GalAse. This is even more evident in case of the Glutamic acid containing form of the Gc protein for which the sialilated form of the protein is very similar in between the untreated and the glycosidases and inhibitor treated protein.
It is also noteworthy that, under those experimental condition, the effect of the inhibitor is marginal since the area of the a-GalNAc linked Gc protein in presence of the inhibitor is less than half of the area in absence of inhibitor.
The panel C requires further analysis, since the only evident effect is over the sialylated form suggesting that the iminocyclitol is an inhibitor of sialidase.
All the discussed results are based on MS/MS exclusively and replica are not shown. Thus, I suggest to use complementary techniques and multiple replica to strengthen the valuable results.
Author Response
Thanks to the reviewer pointing out the important issue about inhibitory effect against GalNAc-ase but also potentially against Sia-ase. We included new evidence of Sia-ase inhibition by the iminicyclitol. We also revised our manuscript by including the new evidence and discussion. (line 226–240) Accordingly, the panels (b) and (c) in Fig. 4 is revised after carrying out some extra experiments.
Quantitative analysis was not possible in our setup based on the ESI-ion trap MS. Furthermore, inaccurate mass intensities over different experiments and handling error associated with the enrichment process of glycopeptides were our major concerns. Instead, we focused on the intensity difference in an experiment to see the tendency of changes.
Osamu Kanie
Round 2
Reviewer 2 Report
The authors have addressed some of my concerns. I still think the sound data based conclusion have not been drawn. For example, in Figure 4, the author did not mention how many replicates were performed and no statistic analysis was performed. If statistic analysis, e.g. T test, was performed, the authors may draw a conclusion if the inhibitory effect is significant different.
Author Response
Thank to the reviewer for the comment. We wish If we could provide more positive reply. We too tried several LS-MS to obtain statistically valid data, but areas obtained in each run differs that we could not perform statistical analysis. However, the LC profiles were similar in each run, so that we decided to shown the data obtained after single run. As for the manuscript, we stated this to show the readers more precisely. Texts added in lines 186–191 are ”Note that the quantitative analysis was not possible in our current analysis setup based on the ESI-ion trap MS. Thus, areas for ions of individual compounds in a mass chromatogram were compared. Our major concern associates with handling errors for the enrichment process of glycopeptides over experiments as well. Instead, we focused on the intensity differences of glycosylated peptides in an individual experiment to see the tendency of changes. (See Fig. S3 for similar experimental data) “. A new figure S3 was included in the supporting material.
Reviewer 3 Report
I am glad that the authors revised the article including the proposed observation that the iminocyclitol inhibit, at least partially, other glycosylhydrolases such as the sialidase. I still belive that the conclusion rised by the authors should be further supported by complementary techniques. Also, the broad inhibitory effect of the iminocyclitol against others glycosylhydrolases, beyond the a-GalNAc-ase, rise concern about the title and the conclusion of the work. Despite those observation, I consider that the work deserve pubblication in Molecules.
Author Response
We revised the title reflecting the reviewer’s suggestion. The new title is “Investigation of the protective effect for GcMAF by a glycosidase inhibitor and the glycan structure of Gc protein ”.